# Exploring the Implementation of GaAsBi Alloys as Strain-Reducing Layers in InAs/GaAs Quantum Dots

**DOI:** 10.3390/nano14040375

**Published:** 2024-02-17

**Authors:** Verónica Braza, Daniel Fernández, Teresa Ben, Sara Flores, Nicholas James Bailey, Matthew Carr, Robert Richards, David Gonzalez

**Affiliations:** 1Institute of Research on Electron Microscopy and Materials (IMEYMAT), The University of Cadiz, 11510 Puerto Real, Spain; veronica.braza@uca.es (V.B.); teresa.ben@uca.es (T.B.); sara.flores@uca.es (S.F.); david.gonzalez@uca.es (D.G.); 2Department of Electronic and Electrical Engineering, The University of Sheffield, Sir Frederick Mappin Building, Mappin Street, Sheffield S1 3JD, UK; nick.bailey@sheffield.ac.uk (N.J.B.); mcarr1@sheffield.ac.uk (M.C.); r.richards@sheffield.ac.uk (R.R.)

**Keywords:** GaAsBi capping, self-assembled InAs QDs, STEM compositional analyses

## Abstract

This paper investigates the effect of GaAsBi strain reduction layers (SRLs) on InAs QDs with different Bi fluxes to achieve nanostructures with improved temperature stability. The SRLs are grown at a lower temperature (370 °C) than the usual capping temperature for InAs QDs (510 °C). The study finds that GaAs capping at low temperatures reduces QD decomposition and leads to larger pyramidal dots but also increases the threading dislocation (TD) density. When adding Bi to the capping layer, a significant reduction in TD density is observed, but unexpected structural changes also occur. Increasing the Bi flux does not increase the Bi content but rather the layer thickness. The maximum Bi content for all layers is 2.4%. A higher Bi flux causes earlier Bi incorporation, along with the formation of an additional InGaAs layer above the GaAsBi layer due to In segregation from QD erosion. Additionally, the implementation of GaAsBi SRLs results in smaller dots due to enhanced QD decomposition, which is contrary to the expected function of an SRL. No droplets were detected on the surface of any sample, but we did observe regions of horizontal nanowires within the epilayers for the Bi-rich samples, indicating nanoparticle formation.

## 1. Introduction

Semiconductor devices based on QDs have shown remarkable advantages due to their zero-dimensional nature, which allows for three-dimensional carrier confinement and a delta-like density of states. Among the QD systems, self-assembled InAs/GaAs QDs grown via the Stranski–Krastanov (S–K) method have attracted much attention for their applications in many fields such as sensor devices [1,2], photovoltaic cells [3,4], and single-photon or entangled photon pairs sources in emerging modern photonic quantum technologies [5,6]. These applications receive help from the high optical activity, low threshold current density, lower dark current, and single-photon emission of InAs/GaAs QDs.

These QDs need to be coated with a passivating capping layer (CL) to protect them from the external environment and reduce nonradiative carrier recombination via surface states. However, the growth parameters of the capping process (temperature, capping rate, flux ratios, and composition) are crucial for their finest performance as they affect the final structural properties of the QDs, such as size, shape, strain fields, and compositional distribution due to a significant material redistribution [7,8]. Therefore, the final nanostructure may differ from the theoretical design if a suitable homogeneous defect-free composition is not achieved.

For this CL, the first choice is to use the same material of the substrate, GaAs, and typical GaAs/InAs/GaAs QDs have an emission wavelength of 1100 nm at 77 K [9]. However, to reach the telecom O (1260–1360 nm) and C bands (1530–1565 nm), the operating wavelength can be extended by using a material different from GaAs to cover the InAs QDs. A common strategy is to apply a strain reduction layer (SRL) made of a material that should have a larger lattice constant than GaAs, which can reduce the QD hydrostatic strain, increasing their emission wavelength. The first SRL material used for InAs QD lasers was InGaAs [10,11], but other SRL materials have also been explored, such as InAlAs [12], GaAsSb [13,14], and quaternary compounds such as InGaAsN [15,16], InGaAsSb [17], GaAsSbN [18,19], or GaAsSb/GaAsN superlattices [20]. These SRLs can offer more flexibility in band structure design and other advantages such as avoiding In segregation [21], increasing the energy separation between the ground and excited states [22], suppressing QD decomposition [23], forming type-II band structures [24,25], and overall, extending the emission wavelength [7,9].

Therefore, it is important to study the effects of novel SRL materials on InAs QDs. One of the potential candidates is the diluted GaAs_1−*x*_Bi*_x_* system, which has a larger lattice constant than GaAs, making it suitable for extending the emission wavelength of InAs QDs [26]. In addition, this dilute bismuthide material opens new avenues for the design of novel devices due to its large bandgap bowing effect, temperature-insensitive bandgap, and large spin-orbit splitting [27]. First, GaAsBi produces a strong and approximately linear bandgap reduction at a rate of 64 meV/%Bi. This value is higher than the bandgap reduction rate of InGaAs (16 meV/%In) and GaAsSb alloying (21 meV/%Sb) but lower than that of nitrogen (125 meV/%N). However, the latter can show degraded optical response due to N/As interstitial defects [28,29]. Second, GaAsBi presents a giant bowing of the spin-orbit splitting energy (*E*_SO_) in the valence band, which offers enormous potential to improve both device performance and thermal stability at high temperatures in the near and mid-infrared wavelengths [30].

However, the achievement of high Bi concentrations in III-V alloys is a challenge due to a large miscibility gap and a low equilibrium solubility of the solid (*x* < 5.2 × 10^−5^) [31] requiring nonequilibrium growth techniques such as molecular beam epitaxy (MBE) and metalorganic vapor phase epitaxy (MOVPE) to access these unique bandgap/lattice constant combinations. Even then, the alloy is prone to develop compositional defects during epitaxial growth, such as clustering, phase separation, Bi droplet formation, atomic ordering and/or strong surface segregation [32,33]. Consequently, Bi incorporation is extremely sensitive to growth conditions, and this poses an added challenge for using it as an SRL. On the one hand, the overgrowth temperature of a CL for InAs QDs should be close to or slightly lower than the QD growth temperature to avoid excessive intermixing and erosion of the QD apexes [34,35]. However, at typical GaAs growth temperatures (~580 °C), Bi can only be used as a surfactant since it segregates rather than incorporates [36,37]. GaAsBi layers are only achievable at low temperatures below ~400 °C. On the other hand, capping InAs QDs with GaAs grown at low temperatures (300 °C), despite preserving sharp QD pyramidal shapes and planar interfaces, results in poor structural and optical quality [38,39]. Although these conditions may seem incompatible, the addition of Bi to an SRL in this low-temperature regime may provide surprises. For example, it has been reported that the addition of Bi during the growth of InAs/GaAs (001) QDs at low temperatures promotes the formation of QDs that are not observed without Bi. In fact, the Bi flux leads to a rapid increase in QD size [36].

All these reasons explain the lack of a systematic study of this system. To the best of our knowledge, Wang et al. [40] were the first to demonstrate the concept of using GaAsBi as an SRL on InAs QDs, achieving a redshift of 10 meV by growing the GaAsBi layer at 400 °C. By lowering the growth temperature to 280 °C, the same group demonstrated a 163 meV PL spectral redshift, resulting in a 1.365 μm emission wavelength at 77 K but without achieving a signal at room temperature [26]. These remarkable studies demonstrate the potential of the system and invite further research to understand how the distribution of Bi occurs when it acts as an SRL.

In this article, we investigate the compositional distribution that occurs during the implementation of GaAsBi SRLs using different Bi fluxes on InAs QDs to obtain a new, less temperature-sensitive nanostructured system. For this, we have set the growth temperature of GaAsBi SRL at 370 °C, low enough to incorporate Bi but far from standard capping temperatures (~500 °C). An exhaustive characterization of the GaAsBi/InAs QDs system has been carried out using different (scanning) transmission electron microscopy ((S)TEM) techniques. As we will see, the effects on the system are vastly different from those obtained when Bi was introduced during the growth of InAs QDs.

## 2. Materials and Methods

Five samples were grown using solid-state molecular beam epitaxy (MBE) on GaAs (001) n^+^ substrates under As_4_ overpressure conditions. First, the native oxide was removed at 625 °C under an As flux, followed by a 300 nm undoped buffer of GaAs grown at a temperature of 580 °C and a growth rate of ~0.3 µ/h and an As_2_:Ga atomic flux ratio of ~1.6. InAs QDs were then deposited at a temperature of 510 °C up to a thickness of 2.2 ML in an S–K growth mode. The growth rate of the QDs was 0.0096 ML/s, with an As_4_:In atomic flux ratio of approximately 36. After the first layer of QDs, the heater power supply was cut off for 60 s. The target temperature was then set at 370 °C, and CL growth started 60 s later. Thus, a 10 nm CL of GaAs(Bi) was deposited under a stoichiometric As_4_ flux at a growth rate of 0.3 µ/h, using 4 different Bi fluxes, ranging from 0 in the reference sample (Ref-LT) up to 2.48 nA in the richest Bi content sample (see Table 1). An additional control sample was created, where the capping layer included a 10 nm GaAs layer grown at 510 °C (Ref-HT). For all samples, a 50 nm GaAs spacer was grown at the same temperature as the CL, with an As_4_:Ga atomic flux ratio of ~2.1. Finally, a new layer of QDs was deposited on the surface using the same protocol as the earlier QD layer. This layer of QDs was left uncapped (see Figure 1). 

Cross-sectional samples were prepared using the lamella method on a Focused Ion Beam microscope (FEI Scios™ 2 Dual Beam™) (Thermo Fisher, Waltham, MA, USA) or by mechanical thinning followed by ion polishing. Structural (contrast diffraction (DC) and annular dark-field (ADF) imaging) and spectroscopic (electron energy loss spectroscopy (EELS) and energy-dispersive X-ray spectroscopy (EDX)) studies were carried out in a Talos F200X (Thermo Fisher) and a double aberration-corrected FEI Titan^3^ Themis microscope (FEI, Hillsboro, OR, USA), both used at 200 kV. EDX maps were obtained by using ChemiSTEM^®^ technology with four integrated Bruker SDD (Silicon Drift Detectors) detectors and processed using Velox^®^ software (version 3.5.0.952).

## 3. Results and Discussion

The properties of the QDs are influenced by the capping process in a complicated way. The capping process not only changes the energy band offsets and strain fields in the QDs but also modifies the size, shape, and composition of the QDs through intermixing and migration processes [41,42]. Indeed, the final structure of the QDs depends on the surface and interface diffusion factors, which are related to the CL growth temperature by an Arrhenius dependence and on the lattice mismatch with the CL. Therefore, the growth temperature of the CL is a key factor for studying how GaAsBi SRLs influence the evolution of InAs QDs. For this study, we selected 370 °C as the best temperature to achieve a good Bi incorporation, below the Bi evaporation temperature in GaAs, which is around 420 °C, but high enough to prevent a high density of shallow and deep electronic states in the bandgap, which could degrade the quality of the structure [43,44]. However, this temperature is well below that of the standard used to cap with GaAs, which is around 500 °C.

To differentiate and understand the effects of CL growth at low temperatures due to the impact of the presence of Bi, two reference samples were grown using CLs without Bi. In the first, sample Ref-HT, the CL was grown at a temperature of 510 °C, which is the standard growth temperature of the QDs, and in the second, sample Ref-LT, the GaAs CL was grown at a lower temperature (370 °C), which is the same temperature used in the samples with GaAsBi SRLs. Figure 2a shows a cross-sectional view along the <110> of both samples using ADF conditions. In the case of surface dots, their shapes stand out against the organic glue background, showing a pyramidal shape. The buried dots appear as islands over a continuous fringe named the wetting layer (WL), typical of the S-K growth mode, by presenting a brighter contrast than the matrix because of the higher In content. In the case of the Ref-HT sample, QDs grow pseudomorphically on the substrate without the presence of crystalline defects. They are smaller than their superficial counterparts, have a lenticular shape, and a lower superficial density [45]. In contrast, in the reference sample with a GaAs CL grown at a low temperature, Ref-LT, buried QDs are larger than the ones grown at a high-temperature CL, showing a pyramidal shape, but also present a high density of threading dislocations (TDs), which are distinguished in the images as bright lines arising from the dots. The presence of these defects is associated with exceeding the critical thickness for plastic relaxation in the QDs, inducing the formation of misfit dislocations to relieve the elastic strain inside the QDs that thread towards the surface.

To quantify the change in QD size, several cross-sectional ADF images taken along both <110> were used to obtain a statistical distribution of the base diameters and heights of the buried QDs. The QDs of all samples were measured using the same criteria, with the height being the vertical distance from the QD tip to the GaAs substrate and the base diameter being the horizontal width of the QDs just above the WL. Figure 3 shows the average QD height and base diameter of all samples. As can be seen, QDs of the sample Ref-LT have a larger base diameter and, overall, an almost double QD height compared to those grown at elevated temperatures. These results suggest that low-temperature capping reduces the decomposition of QDs compared to high-temperature capping so that the resulting capped QDs are larger and closer in shape to the bare QDs of the surface. However, many of these QDs are so large that plastic relaxation of the strain energy is possible through the formation of misfit dislocations, which degrades the optical performance of the material. It seems that a certain amount of erosion of QDs, such as that obtained during high-temperature GaAs capping, is needed for buried QDs to reach a suitable size and content that does not exceed the critical thickness for plastic relaxation.

Similar analyses were conducted for the samples with Bi in the CL. First, surface (not shown here) and cross-sectional analyses showed no evidence of Bi-rich droplet formation in any of them. Starting with the sample with the lowest Bi content, Bi-L, Figure 2b shows a horizontal shiny layer about 9 nm thick over the dots, which we presume to be the GaAsBi layer. This layer does not appear immediately after the WL but shows a gap between them. The strong tendency of Bi to segregate toward the surface results in a spatial separation between the WL and the SRL, although this separation does not occur in the regions around the QDs, where their apex appears to be in contact with the GaAsBi SRL. This regular scenario changes in the samples with the higher Bi flux, Bi-M, and Bi-H, where two distinct types of alternating regions can be distinguished. In one, which we call Regular Zones (RZ)—visible in Figure 2c,e for samples Bi-M and Bi-H, respectively—QDs appear to be covered by a brighter layer thicker than that seen in the sample Bi-L. In the other zones, named Highly Segregated Zones (HSZ), as found in Figure 2d,f for samples Bi-M and Bi-H, respectively, the GaAsBi layer has been displaced upwards from its nominal position. In these regions, the number of QDs in these areas gradually decreases, and, in certain areas, both the WL and the InAs dots even disappear, as can be seen in the center of Figure 2d for the Bi-M sample. The proportion of these HSZ is higher with increasing Bi flux. In the first place, we will concentrate on the study of the RZ, which we consider the most representative of the material, dedicating the last part of the section to the HSZ and the possible causes of their formation.

In the RZ, GaAsBi layers keep a constant thickness by adapting to the topology of the underlying QD layer. Notably, as seen in Figure 2c, a significant decrease in the crystalline defect density compared to Ref-LT is appreciated that is linked to an important reduction in QD size. As can be seen in Figure 3, the average QD height falls from 10.3 to 8.2 nm in the Bi-L sample, although it experiences a small upturn as Bi fluxes increase. The effect of Bi flux on the base diameter of the QDs is opposite to that of the height. The base diameter gradually decreases from 29 nm in Ref-LT to 24 nm in Bi-H with increasing Bi flux. This implies that the QD volume also decreases with the Bi flux since the base diameter is the main factor in determining the QD volume. In the low-temperature regime, the increased Bi flux decreases the protective effect of the SRL, inducing a greater QD decomposition. This behavior is different from what occurs when Bi is used as a surfactant during the growth of InAs QDs at low temperatures, which produces an increase of both QD height and base diameters [36].

Due to the complexity of categorizing the different contrasts in ADF images, EDX compositional maps of the different areas were carried out. Focusing on the RZ, Figure 4 shows the (a) colormaps of representative regions of all the samples grown at LT, where In and Bi distribution is hued in green and red, respectively, and (b) the average composition profiles along the growth direction of the areas marked with arrows. The width of the areas for the profile quantification is 30 nm. As shown in the figure, the thickness of the GaAsBi SRL increases with Bi flux, adapting its shape to the morphology of the buried InAs QD layer. The Bi content reaches a maximum value of 2.4% in all samples and the excess of the Bi supply seems to incorporate in the epitaxy by increasing the layer thickness. Figure 5 shows the area under the Bi profile along the growth direction, which is proportional to the amount of Bi introduced during the epitaxy growth versus Bi fluxes. The area under the profiles evolves almost linearly in contrast to the observed saturation of the maximum Bi content with increasing Bi flux. Regardless of the increase in Bi flux, the Bi content reaches the solubility limit for GaAs at this growth temperature in MBE, which is about 2.4% [33], and the excess of Bi accumulates in the floating layer and is slowly incorporated as the CL layer grows. In addition, the homogeneity of the compositional distribution of Bi changes when compared among the samples. The Bi-L sample shows a high uniformity of the Bi content along the growth plane that is relatively preserved in the Bi-M sample. However, in the Bi-H sample, Bi is preferentially accumulated in the zones between the dots, decreasing as we go up the layer. Furthermore, the position of the start of the GaAsBi layer differs. Thus, in the profiles of the Bi-L sample, a gap between the WL and the GaAsBi CL is noticed. This gap between the In peak of the WL and the steady state content of the Bi profile linearly shortens as the flux increases and almost disappears in the Bi-H sample, as can be seen in the profiles of Figure 4b. Increasing the Bi flux modifies the growth front, allowing the Bi to be incorporated earlier.

GaAsBi CL deposition can also affect the distribution of In in the InAs QD layer. In other words, the amount of In present in the QDs and the WL can shift. Due to this interconnection, analysis of the WL can provide important clues about what happens during QD formation, as it undergoes significant changes throughout the growth process. First, at the beginning of epitaxial growth of InAs on GaAs substrates, the WL thickness increases linearly with the amount of deposited InAs [46]. When the critical thickness for the transition to the S-K growth mode is reached, island formation occurs, and a transfer of material from the WL to the newborn QDs takes place to sustain the development of the QDs [47,48]. This process leads to a halt in WL growth or even a reduction in WL thickness [49]. However, this material transfer can be reversed during the capping process, where the system tends to reduce the total energy by relaxing the local deformation within the QD by segregating In from the QDs to the WL [50,51]. The larger the decomposition of the QDs, the greater the thickness of the resulting WL.

To analyze the changes in the WL, Figure 4b plots the In profiles of regions between QDs as green lines. As can be seen, the In peaks corresponding to the WL become narrower as the Bi flux increases, implying that the segregation of In is influenced by the presence of Bi during capping. Certainly, the slopes of the In profiles are higher in both the up and down regions. First, at the beginning of CL deposition, the floating layer responsible for the surface segregation is mainly occupied by Bi atoms since the segregation energy of Bi/As (0.07 eV) [52] is much lower than that of In/Ga (0.2 eV) [53]. In atoms tend to be incorporated into the film more rapidly in the presence of Bi than with Ga alone, and the composition gradient in the upward region of the In profiles is greater. Second, the reason for the faster decay after the maxima is more subtle. As Eisele et al. pointed out [54], the back-segregation of In coming from QDs decomposition critically affects the resulting In profile of the WL, but only in the downward region [51,55]. Given this, the rapid decay of the segregation tail in the Bi samples should be related to a decreased decomposition rate of the InAs QDs, but we saw the opposite when comparing the evolution of the QD sizes in Figure 3.

The general hypothesis is that there is a direct relationship between the degree of degradation of the QDs and the amount of In in the WL: the thicker the WL, the smaller the size of the QDs. However, we do not know whether this decrease in size also implies changes in the average In content inside the QDs [51,56]. To evaluate the latter, we have measured the In content of several individual QDs from EDX maps considering both the QD size and the TEM sample thickness using low-loss EELS maps [23]. The results of the average In content for a substantial population of QDs in all samples grown at LT are remarkably similar in all cases, ranging from 70–80%. Remember that the original population of InAs QDs is the same in all the samples, and only the amount of Bi present during capping is changed. It seems that the capping process with Bi only leads to a loss of InAs from the QD linked to size reduction but not to a dilution of the In content by a significant Ga/In exchange process.

Taking all of this into account, the areas under the In profiles of the WL regions (black squares in Figure 5) and the QD sizes (Figure 3) allow us to evaluate the QD decomposition of the different SRLs [51]. In the case of the SRL with the lowest Bi content, sample Bi-L, the WL area is larger than the one of the sample Ref-LT, which is explained by the meaningful volume decrease of the QDs. For the Bi-L and Bi-H samples, the reduction in QD volume compared to the Ref-LT sample is even higher, at approximately 30% and 40%, respectively. One would expect a large increase in WL area in the more Bi-rich samples due to the significant decrease in QD volume, but this does not happen, as can be seen in Figure 5.

The results of the analysis of the QD sizes and the WL area led us to a situation that seems contradictory for the samples richer in Bi. However, the explanation for the latter can be found in the more detailed analysis of the In profiles in Figure 4, where the presence of an unexpected In signal above the GaAsBi layer can be detected. This additional InGaAs layer becomes more distant from the InAs QDs layer as the GaAsBi layer thickens. Notably, the In signal is detected 50 nm above the WL in the Bi-H sample. The maximum In content never exceeds 3%, although the layer lengthens with increasing Bi flux. The amount of In deposited on top of the GaAsBi layer is much higher in the Bi-H and Bi-M samples than in the Bi-L sample, as can be seen in Figure 5, which represents the total area under the In profile segregated above the GaAsBi layer. We believe that the In content of this segregated InGaAs layer atop the GaAsBi CL is derived from the decomposition of the QDs during capping, where Bi pulls In atoms toward the growth front and separates them from the WL. Note that the distribution of In in this segregated layer is not homogeneous and preferentially accumulates above the underlying QDs. This effect may be related to the earlier incorporation of Bi in these samples. In fact, the distance between the WL and the SRL decreases linearly from 8 nm in the Bi-L sample to 4 nm in the Bi-H sample. In the case of the Bi-L sample, the lower Bi content in the floating layer allows the In atoms from the QD decomposition not to be dragged to the surface and incorporated into the WL, so the amount of In segregated in the GaAsBi layer is minimal.

Although there are numerous studies in the literature on how the use of a compressive CL limits the erosion of InAs/GaAs QDs [13,14], we have shown that at these low growth temperatures, the presence of Bi does not provide a shielding effect against QD erosion, quite the opposite. However, a more detailed study of the SRL behavior shows that this is not always the case when we move away from the standard GaAs capping conditions. It has been described that at higher growth rates [8] or at lower growth temperatures [41] for GaAsSb, SRLs do not prevent QD degradation but produce smaller InAs QDs than GaAs CL samples. These results suggest that the role of SRLs in the capping process is more complex and not always effective in protecting the integrity of InAs QDs, especially in the case of species with strong surfactant-like effects.

Unfortunately, in samples with higher Bi content, defective regions, which we have termed HSZ, appear between the RZ regions, the extent of which increases with Bi flux. Figure 6a shows an extended DCTEM g002DF image of one of these zones. For these compositions, the brightest and darkest contrasts in the image correspond to Bi and In-rich regions, respectively. In these regions, the GaAsBi layer appears far away from the InAs QD layer, sometimes as much as 20–30 nm above the InAs QD layer. EDX analysis on these HSZs for the Bi-M and Bi-H samples (see Figure 6b and Figure 6c, respectively) confirm the complete absence of Bi in the channel between the CL and the WL, as well as the disappearance of In in certain areas of the InAs QDs layer. The GaAsBi layers are thinner than those corresponding to the RZ but have the same maximum content of around 2.4% Bi. Triangular nanostructures are also observed above the WL, sometimes coinciding with QDs (see Figure 6b,c). The analysis of the triangular nanostructures showed that they have the same Bi content as the CL, so we assume that they are structures extended in the direction of the electron beam, like a nanowire, instead of an island. Notably, these GaAsBi layers have flat interfaces, but they are not always parallel to the growth plane, with angles of 5–8°, as is shown in Figure 6c. The presence of these free Bi channels has been linked to the strong predisposition of Bi to form droplets during the growth of Bi-rich alloys [57]. This tendency is strongly dependent on the Bi/As flux ratio, making the transition very sharp [58]. This agrees with the observation of HSZs in our samples with a high Bi flux ratio and their complete absence in the sample with a low flux ratio. The Bi droplets are very mobile and roll almost perpendicular to the growth direction due to the low strength of the Ga-Bi bond. This leads to the formation of the experimentally observed GaAs channels as the droplet sweeps Bi away from the underlying film [59,60], which, in turn, can pull some of the In out of the InAs layer. The absence of Bi-rich droplets on the surface of these samples is related to possible evaporation of it during the growth of surface QDs (510 °C). In general, Bi droplets do not etch GaAs, so we have no unambiguous evidence of their existence if desorption of them has occurred [57].

These results represent the first step towards the realization of GaAsBi/InAs/GaAs QDs nanostructures. First, we have seen that the growth of the CL at low temperatures results in a significant decrease in the erosion of QDs, producing very large QDs that plastically relax, causing TDs that degrade the optical properties. This problem could be avoided by reducing the ML number of the InAs layer so that the bare QDs are already smaller to begin with. Furthermore, it has already been shown that the presence of Bi can stimulate the formation of QDs at these temperatures [36]. Progress has also been made on the conditions required for the GaAsBi SRL. We can control the thickness of the layer, but the maximum content is limited to 2.4%. High Bi fluxes should be avoided as they cause undesirable segregation phenomena that destroy the structure. Further work is underway to study the formation of these defect-free nanostructures.

## 4. Conclusions

In summary, the possibility of introducing GaAsBi SRLs in the development of new InAs/GaAs QDs devices using different Bi fluxes has been explored. Since its application requires growth at lower temperatures than those commonly used for standard InAs QD coating, we have separately analyzed the effect of CL growth at this temperature. The use of GaAs capping at low temperatures showed a reduction in dot decomposition with larger pyramidal dots compared to those studied at the typical growth temperature (510 °C), resulting in the formation of a high TD density. The addition of Bi in the CL shows a significant decrease in the density of TDs, but at the expense of the occurrence of significant structural changes that were not expected. Firstly, all GaAsBi layers reach the solubility limit (2.4% Bi) while keeping a constant thickness that matches the topology of the QD layer. At higher fluxes, the excess Bi does not lead to an increase in Bi content but to an increase in layer thickness, bringing forward its effective incorporation into the epitaxy, with a decrease in the distance between the wetting layer and the capping layer. Second, the use of GaAsBi SRLs does not protect the InAs QDs from degradation; on the contrary, a significant reduction in QD size (up to 40%) is seen with Bi flux compared to the reference sample at low temperatures. The standard transfer of In segregated from the decomposition of the QDs during capping to the wetting layer only occurs in the low Bi flux sample. Surprisingly, this does not occur in the more Bi-rich samples, where an extra InGaAs layer is formed on top of the GaAsBi layer due to the Bi carrying away the In atoms resulting from the decomposition of the QDs. Unfortunately, although no droplets were seen on the surface, the increased Bi flux leads to the formation of regions of high segregation, which are attributed to the lateral movement of the Bi-rich droplets that disappeared during the final temperature rise.

## Figures and Tables

**Figure 1 nanomaterials-14-00375-f001:**
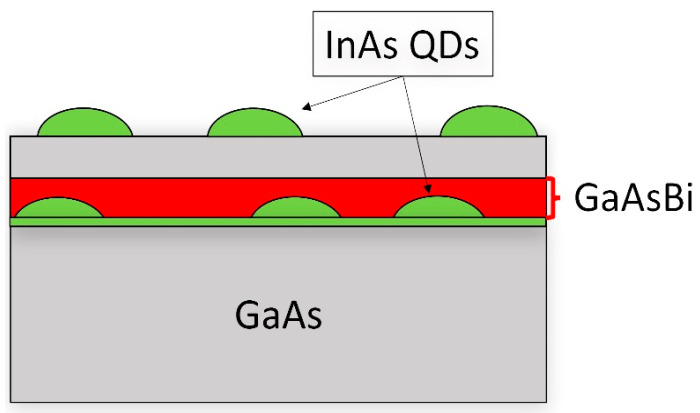
Simple scheme of the samples. Two InAs QD layers are grown. The buried QD layer is capped by a 10 nm GaAs(Bi) layer plus 40 nm GaAs layer at the same temperature.

**Figure 2 nanomaterials-14-00375-f002:**
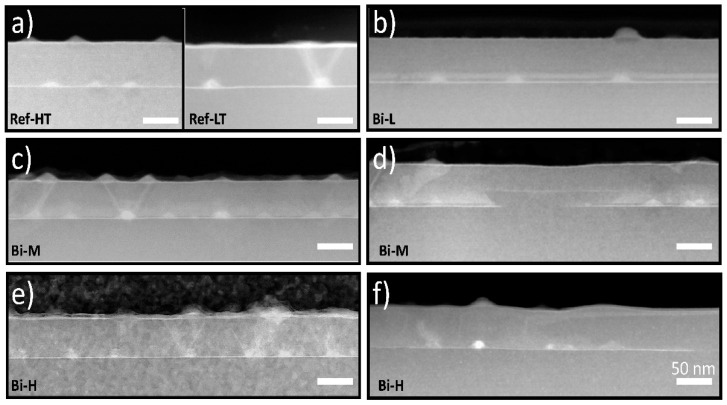
(**a**–**f**) ADF images of all QD samples along <110> at low magnification. The scale bar is 50 nm for all images, and the name of each sample is shown in the lower left corner. In the case of samples with higher Bi flux, Bi-M and Bi-H, we present images corresponding to the two distinct regions found in these samples. On the left, (images (**c**,**e**)), Regular Zones (RZ) are shown where QDs appear to be covered by a homogeneous GaAsBi layer. On the right, (images (**d**,**f**)) in the so-called Highly Segregated Zones (HSZ), the GaAsBi layer has moved upwards with respect to its nominal position.

**Figure 3 nanomaterials-14-00375-f003:**
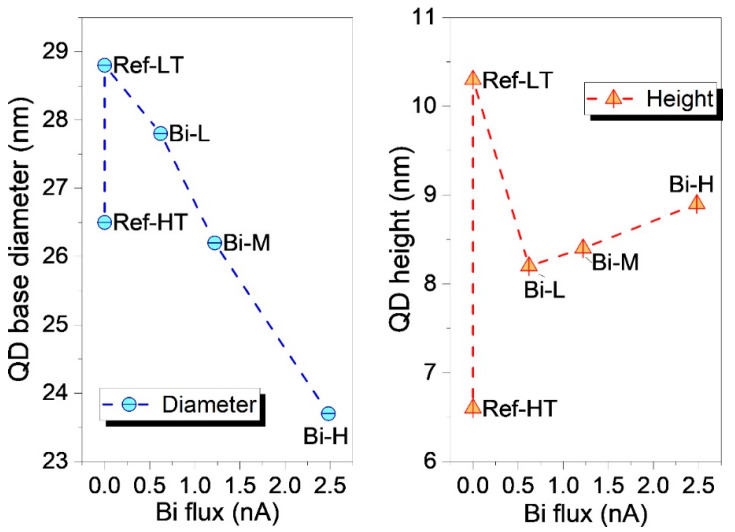
Plots of the average heights (**right**) and base diameters (**left**) of buried QDs of all the samples.

**Figure 4 nanomaterials-14-00375-f004:**
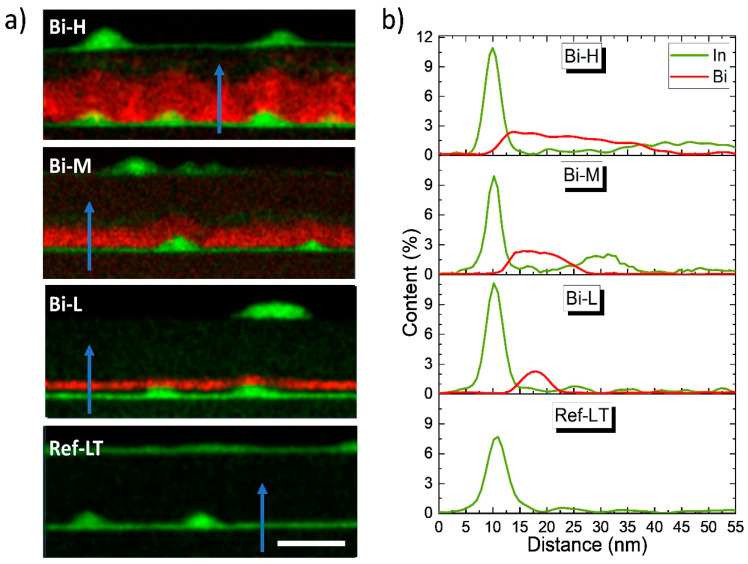
(**a**) EDX color maps for In (green) and Bi (red) distribution taken in the RZ. (**b**) Average compositional profiles of Bi and In in the areas between the QDs, indicated by the blue arrows. The scale bar is 50 nm. The grey squares inside the profiles mark the separation between WL and the GaAsBi layer.

**Figure 5 nanomaterials-14-00375-f005:**
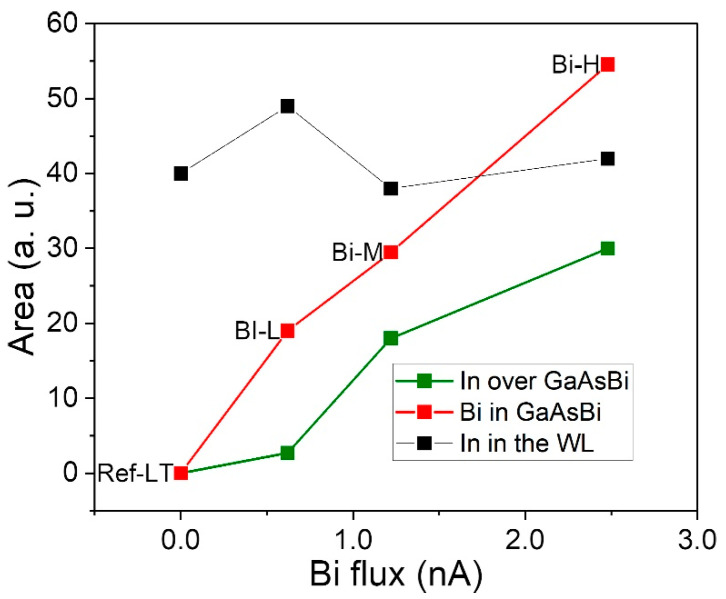
Area under of the Bi profile of the CL (red) and of the In profile incorporated into the WL (black) and after the GaAsBi layer (green) along the growth direction versus the Bi flux.

**Figure 6 nanomaterials-14-00375-f006:**
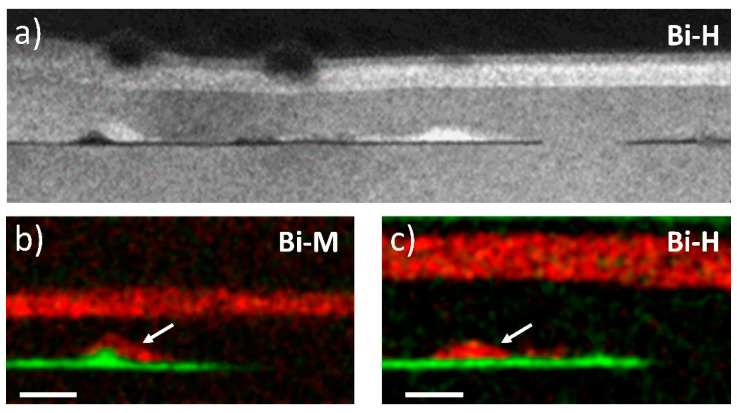
(**a**) DCTEM using g002 DF conditions of HSZ region of Bi-H sample. Brighter and darker contrasts correspond to Bi and In-rich regions, respectively. EDX maps for In (green) and Bi (red) distribution of HSZ regions of sample Bi-M (**b**) and Bi-H (**c**). Arrows mark triangular nanostructures. The scale bar is 20 nm.

**Table 1 nanomaterials-14-00375-t001:** The nomenclature of the samples based on the growth conditions of the CL.

Sample	Bi Flux (nA)	Growth Temperature (°C)
Ref-HT	0	510
Ref-LT	0	370
Bi-L	0.62	
Bi-M	1.22	
Bi-H	2.48	

## Data Availability

The data that support the findings of this study are available from the corresponding author upon reasonable request.

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
