# Peer review of "Exploring the Implementation of GaAsBi Alloys as Strain-Reducing Layers in InAs/GaAs Quantum Dots"

_nanomaterials, 2024, doi:10.3390/nano14040375_

Round 1
Reviewer 1 Report
Comments and Suggestions for Authors
Braza et al. investigated the effect of GaAsBi SRLs on InAs QDs with different Bi fluxes to achieve nanostructures with improved temperature stability. However, in previous investigations such as DOI 10.1186/s11671-016-1470-1, https://doi.org/10.1016/j.jlumin.2022.119155, https://doi.org/10.1364/OME.7.004249 and many more researcher have investigated similar results. Therefore, I cannot recommend the publication of the manuscript in Nanomaterials.
Comments on the Quality of English LanguageModerate editing of English language required
Author Response
We thank the reviewer for their feedback and have incorporated two of the references they suggested into our paper. However, we do not agree with the claim that there are many other related studies. To the best of our knowledge, the papers that the reviewer suggested are the only ones to explore the application of GaAsBi layers as SRL on InAs dots, and we would appreciate additional examples if there are any. The two papers, from the same research group, examine the surface quality and PL of InAs QDs on GaAs with Bi in the pre- or post-QD layer or both. They demonstrate that a GaAsBi capping layer can effectively lower the bandgap energy and they achieve an emission wavelength record of 1.365 μm at 77 K. This is an impressive result, and we agree that it should be included in the introductory section of our paper.
These articles, however, do not provide a comprehensive analysis of the nanoscale alloy distribution and its effects. Only one of them performed EDX mapping measurements of the surface and capped InAs QDs — and only for the In element without investigating the Bi distribution or incorporation. Furthermore, they did not conduct a detailed study of the problems associated with In and Bi arrangement in both the QDs and the SRL when integrating the low-temperature growth of GaAsBi layers with InAs/GaAs QDs. Hence, a thorough nanoscopic structural, morphological, and compositional analysis is essential to fully understand this system. Our study not only offers this important characterization, but also extends it by examining different Bi fluxes to present a more complete picture of this system.
We acknowledge that we overlooked these relevant previous studies in our initial manuscript, and we apologize for this omission. We agree that these studies should be cited in our paper, as they provide important context and support for our research and findings. Therefore, we have added a new paragraph in the Introduction section, where we review these studies and highlight their contributions. This paragraph is inserted before the last paragraph of the Introduction, where we discuss the state of the art of this system and propose our objectives.
All these reasons explain the lack of a systematic study of this system. To the best of our knowledge, Wang et al. (P. Wang et al., 2016) were the first to demonstrate the concept of using GaAsBi as an SRL on InAs QDs, achieving a redshift of 10 meV by growing the GaAsBi layer at 400 °C. By lowering the growth temperature to 280°C, the same group demonstrated a 163 meV PL spectral redshift, resulting in a 1.365 μm emission wavelength at 77 K, without achieving signal at RT (L. Wang et al., 2018). These remarkable studies demonstrate the potential of the system and invite further research to understand how the distribution of Bi occurs when it acts as an SRL.
New bibliography
Wang, L., Pan, W., Chen, X., Wu, X., Shao, J., & Wang, S. (2018). Influence of Bi on morphology and optical properties of InAs QDs: publisher’s note. Optical Materials Express, 8(9), 2702. https://doi.org/10.1364/ome.8.002702
Wang, P., Pan, W., Wu, X., Liu, J., Cao, C., Wang, S., & Gong, Q. (2016). Influence of GaAsBi Matrix on Optical and Structural Properties of InAs Quantum Dots. Nanoscale Research Letters, 11(1), 280. https://doi.org/10.1186/s11671-016-1470-1
Comments on the Quality of English Language
Moderate editing of English language required.
We have made every effort to correct any errors in the document, such as spelling mistakes, unusual characters, or abbreviations. We hope that this will improve the readability and clarity of the text and avoid any confusion or misunderstanding among the readers.
Reviewer 2 Report
Comments and Suggestions for Authors
The authors submitted a study dealing with a strain-reducing layer of GaAsBi in InAs/GaAs heterostructure. The topic is quite interesting, and there are novel results. I would like to support it; however, there are several issues to fix prior to the discussion on paper acceptance.
Major issue: Experimental proof of the strain reduction
The GaAsBi layer should reduce the mechanical stress, but there is no proof that it has this effect. As a result, the authors should include some study (such as X-ray diffraction) that can reveal the microstress present in the layer.
Minor issue: Proofreading
Usually I do not comment the English level; however, this manuscript contains typos (e.g. "lineal") and uncommon symbols (e.g. "º" instead of "°"). In addition, please introduce every acronym before using it in the text. The first time you use the term, put the acronym in parentheses after the full term.
There is a real need of proofreading.
Author Response
The concept of SRL dates to the early years of this century (Yen et al., 2000) when InAs QDs began to be capped with InGaAs layers. These early studies demonstrated the possibility of a large redshift when InAs/GaAs QDs are capped with any material with a lattice parameter larger than that of the GaAs substrate, as we exposed in the introductory section. However, the theoretical explanation of why it occurs is much later when Liu et al (Liu et al., 2008) demonstrated, within the framework of the eight-band k×p theory, the reduction of the bandgap due to the presence of a strain-reducing layer by calculating the strain distributions around the QDs using finite element methods. The numerical results showed that indeed the biaxial strain (εbi = 2εzz − εrr − εφ) is strengthened in the InAs QD and that only the hydrostatic strain (εhyd = εzz + εrr + εφ) is reduced. The combined effect of these deformations on the energy levels of electrons and heavy holes explains the redshift of the emission wavelength due to the SRL.
Therefore, this checking would imply to determine the strain tensor's 3 components inside the QD and contrasted with and without SRL to verify the SRL effect. This can only be achieved by applying deformation calculation methods in high resolution column-resolved TEM images, where we should also select QDs with and without SRL with the same composition and size. However, we think that this is not essential, with the reviewer's consent. In the paper, we have demonstrated that the SRL enveloping the QDs has Bi with an average concentration of 2.4%, and it is widely acknowledged that the alloy's lattice constants augment with the addition of Bi (Oe, 2002). In our case, it is common to assume compliance with Vegard's law with the hypothetical GaBi lattice constant of 6.33 Å (Lewis et al., 2012), so the SRL lattice constant is approximately 5.67 Å, (aGaAs= 5.6532 Å). The compressive effect is small but significant. In this sense, we have included in the penultimate paragraph of the Introduction section previous works that test the redshift effect of GaAsBi used as SRL.
All these reasons explain the lack of a systematic study of this system. To the best of our knowledge, Wang et al. (P. Wang et al., 2016) were the first to demonstrate the concept of using GaAsBi as an SRL on InAs QDs, achieving a redshift of 10 meV by growing a GaAsBi layer at 400 °C. By lowering the growth temperature to 280 °C, the same group demonstrated a 163 meV PL spectral redshift, resulting in a 1.365 μm emission wavelength at 77 K, without achieving signal at RT (L. Wang et al., 2018). These remarkable studies demonstrate the potential of the system and invite further research to understand how the distribution of Bi occurs when it acts as an SRL.
Minor issue: Proofreading
Usually I do not comment the English level; however, this manuscript contains typos (e.g. "lineal") and uncommon symbols (e.g. "º" instead of "°"). In addition, please introduce every acronym before using it in the text. The first time you use the term, put the acronym in parentheses after the full term.
We have made every effort to correct any errors in the document, such as spelling mistakes, unusual characters, or abbreviations. We hope that this will improve the readability and clarity of the text and avoid any confusion or misunderstanding among the readers.
Bibliography
Lewis, R. B., Masnadi-Shirazi, M., & Tiedje, T. (2012). Growth of high Bi concentration GaAs 1-xBi x by molecular beam epitaxy. Applied Physics Letters, 101(8), 1–5. https://doi.org/10.1063/1.4748172
Liu, Y. M., Yu, Z. Y., & Ren, X. M. (2008). Influence of strain-reducing layer on strain distribution of self-organized InAs/GaAs quantum dot and redshift of photoluminescence wavelength. Chinese Physics Letters, 25(5), 1850–1853. https://doi.org/10.1088/0256-307X/25/5/089
Oe, K. (2002). Characteristics of semiconductor alloy GaAs1-xBix. Japanese Journal of Applied Physics, Part 1: Regular Papers and Short Notes and Review Papers, 41(5 A), 2801–2806. https://doi.org/10.1143/JJAP.41.2801
Wang, L., Pan, W., Chen, X., Wu, X., Shao, J., & Wang, S. (2018). Influence of Bi on morphology and optical properties of InAs QDs: publisher’s note. Optical Materials Express, 8(9), 2702. https://doi.org/10.1364/ome.8.002702
Wang, P., Pan, W., Wu, X., Liu, J., Cao, C., Wang, S., & Gong, Q. (2016). Influence of GaAsBi Matrix on Optical and Structural Properties of InAs Quantum Dots. Nanoscale Research Letters, 11(1), 280. https://doi.org/10.1186/s11671-016-1470-1
Yen, N. T., Nee, T. E., Chyi, J. I., Hsu, T. M., & Huang, C. C. (2000). Matrix dependence of strain-induced wavelength shift in self-assembled InAs quantum-dot heterostructures. Applied Physics Letters, 76(12), 1567–1569. https://doi.org/10.1063/1.126097
Reviewer 3 Report
Comments and Suggestions for Authors
In their manuscript entitled “Exploring the implementation of GaAsBi alloys as strain-reducing layers in InAs/GaAs quantum dots” V. Braza et al. report crystal growth studies with solid-source molecular beam epitaxy dedicated to improve control of the properties and quality of self-assembled InAs quantum dots. As pointed out in the introduction of the manuscript, semiconductor quantum dots are still subject of current research, mainly due to the multitude of technological applications such as single-photon emitters, sensors or photovoltaic devices. Studies contributing to a deeper understanding of the Stranski-Krastanov process, on which the quantum dot formation bases, and developing strategies to improve the control of the process are still of high interest to a specialized community.
The idea behind the present studies was to systematically change the composition of the capping layer on the InAs quantum dots. It is expected that with a strain relaxed capping layer the decomposition of the quantum dots, which is known to occur during the capping process, will be reduced. In previous studies the authors have already studied material compositions such as GaAsSb, GaAsN. In the present study the effect of GaAsBi is investigated, which apart from having a larger lattice constant than GaAs is also of interest because of the surprisingly large spin-orbit splitting energy observed in this material even at very low Bi concentrations. A thoughtfully devised series of samples was carefully examined using scanning transmission electron microscopy.
The interesting and partly surprising findings are clearly reported and convincingly discussed. Hence, I can recommend publication of the work in nanomaterials.
The authors might want to consider the following minor editorial points before publication:
Introduce the meaning of TD (threading dislocations) in the abstract.
Replace “a strong bandgap lineal reduction” by “a strong and linear bandgap reduction“ in the third paragraph of page 2.
Introduce a verb in the sentence on the second line on page 3: GaAsBi layers can(?) only be achieved ...
Part of the caption to Fig. 2 is missing.
Delete a superficial “will be” in the 3rd line of text behind Fig. 2.
It is not clear to the reader what “it” refers to in the sentence “Since the base diameter is the dominant parameter in the calculation of the QD volume, it decreases linearly with increasing Bi.”. In my understanding it would refer to the volume, into which the diameter goes quadratically.
The maximum Bi content is given as 2.4 %. Only in one case on page 11 it is given as 2.7 %. Is it a typo?
Comments on the Quality of English LanguageEnglish is okay, some issues are mentioned at the end of the report.
Author Response
Comments and Suggestions for Authors
In their manuscript entitled “Exploring the implementation of GaAsBi alloys as strain-reducing layers in InAs/GaAs quantum dots” V. Braza et al. report crystal growth studies with solid-source molecular beam epitaxy dedicated to improving control of the properties and quality of self-assembled InAs quantum dots. As pointed out in the introduction of the manuscript, semiconductor quantum dots are still subject of current research, mainly due to the multitude of technological applications such as single-photon emitters, sensors, or photovoltaic devices. Studies contributing to a deeper understanding of the Stranski-Krastanov process, on which the quantum dot formation bases, and developing strategies to improve the control of the process are still of high interest to a specialized community.
The idea behind the present studies was to systematically change the composition of the capping layer on the InAs quantum dots. It is expected that with a strain relaxed capping layer the decomposition of the quantum dots, which is known to occur during the capping process, will be reduced. In previous studies the authors have already studied material compositions such as GaAsSb, GaAsN. In the present study the effect of GaAsBi is investigated, which apart from having a larger lattice constant than GaAs is also of interest because of the surprisingly large spin-orbit splitting energy observed in this material even at very low Bi concentrations. A thoughtfully devised series of samples was carefully examined using scanning transmission electron microscopy.
The interesting and partly surprising findings are clearly reported and convincingly discussed. Hence, I can recommend publication of the work in nanomaterials.
The authors might want to consider the following minor editorial points before publication:
We would like to thank the reviewer for his/her comments and constructive criticism, which we believe have helped to improve the quality of our work. We have changed the article in accordance with your suggestions and have made every effort to correct any errors in the document, such as spelling mistakes, unusual characters, or abbreviations. We hope that this will improve the readability and clarity of the text and avoid any confusion or misunderstanding among the readers.
Introduce the meaning of TD (threading dislocations) in the abstract.
Sorry by the omission. It has been introduced.
Replace “a strong bandgap lineal reduction” by “a strong and linear bandgap reduction“ in the third paragraph of page 2. –
It has been replaced.
Introduce a verb in the sentence on the second line on page 3: GaAsBi layers can(?) only be achieved ...
Thank you for the mistake. We have rephrased the sentence as:
GaAsBi layers are only achievable at low temperatures.
Part of the caption to Fig. 2 is missing.
It must have been a mistake when converting the word file to pdf format. It has been corrected.
Delete a superficial “will be” in the 3rd line of text behind Fig. 2.
It has been deleted.
It is not clear to the reader what “it” refers to in the sentence “Since the base diameter is the dominant parameter in the calculation of the QD volume, it decreases linearly with increasing Bi.”. In my understanding it would refer to the volume, into which the diameter goes quadratically.
The sentence has been rephrased as follows:
The effect of Bi flux on the base diameter of the QDs is opposite to that on the height. The base diameter gradually decreases from 29 nm in Ref-LT to 24 nm in Bi-H with increasing Bi flux. This implies that the QD volume also decreases with the Bi flux, since the base diameter is the main factor in determining the QD volume.
The maximum Bi content is given as 2.4 %. Only in one case on page 11 it is given as 2.7 %. Is it a typo?
Yes, it is a typo and has been corrected.
Comments on the Quality of English Language
English is okay, some issues are mentioned at the end of the report.
Round 2
Reviewer 1 Report
Comments and Suggestions for Authors
The authors have made significant changes, and I believe this manuscript can be accepted in its current form without requiring additional revisions.
Reviewer 2 Report
Comments and Suggestions for Authors
The authors responded to the reviewers' comments and questions. I am not fully satisfied since there are no experimental proofs; however, the references regarding the strain-relasing-layers are suffiecient for now. As a result, I have no objectionand I recommend to accept the mansucript in present form.
Comments on the Quality of English LanguageNo comments on English level.